# Can the Brazilian National Logistics Plan Induce Port Competitiveness by Reshaping the Port Service Areas?

**William Costa [1,*], Britaldo Soares-Filho [1] and Rodrigo Nobrega [2]**

1   Remote Sensing Center, Federal University of Minas Gerais, Belo Horizonte 31270-901, Brazil
2   Transportation Research and Environmental Modelling Lab, Institute of Geosciences,
    Federal University of Minas Gerais, Belo Horizonte 31270-901, Brazil
*   Correspondence: williamleles@gmail.com; Tel.: +55-31-3409-7512

**Abstract:** Brazil's transportation infrastructure did not follow the country's agricultural development and the macro-logistics operations still rely on trucking. Even with a lack of roads, the service areas of the ports on the Atlantic coast, particularly the port of Santos, expanded to central Brazil, the country's most productive agricultural area. Recently, the Federal Government released mid-term plans to build railways to reduce transportation costs until 2035. However, no simulation about port regionalization and competitiveness was performed. This research evaluated the effectiveness of the proposed transportation infrastructure regarding transportation cost and new routes that can reshape the ports' influence areas. Our geographically explicit model used the Dinamica_EGO modeling platform and PostgreSQL, fed by official public data from transportation and agriculture authorities. Considering the hypothesis that new railways can increase port competitiveness, we computed scenarios considering the planned 2035 infrastructure and compared them to the current situation. The findings showed that the Ferrogrão railway can effectively reduce transport costs, therefore changing the spatial configuration of macro-logistics basins. In conclusion, a geospatial model can predict short-cheaper routes, port regionalization, and competitiveness regarding the geographic aspects of the supply chain. The long-distance and importance of Brazilian agriculture exportation justify and value the investigation.

**Keywords:** transportation cost; Brazilian ports; port competitiveness; agriculture commodity; macro-logistics

## 1. Introduction

The importance of ports for a country's trade and economic development is undeniable. According to data from the Statistical Yearbook produced by the Brazilian National Waterway Transport Agency—ANTAQ, the port sector represents 95% of the foreign trade flow in Brazil. In addition, the revenue of the port system reaches an average of 293 billion reais annually, which represents 14.2% of the Brazilian GDP and is responsible for generating 120,000 direct and indirect jobs. However, it is essential to highlight that, unlike in highly industrialized countries of Asia [1,2], Europe [3–5], and North America [6,7], the Brazilian ports are generally not geographically aggregated with industries. On the contrary, many products that enter the country through the ports are moved by highways or rails to the industries and consumers' places [8]. On average, export products, mostly primary minerals and agricultural commodities, travel hundreds of kilometers between the origin and destination port [8–10].

In addition to Brazil's infrastructure being in relatively good shape compared to other South American countries [11], the expansion of agricultural production in the interior of Brazil and the intensification of large-scale mineral production experienced at the end of the 20th century revealed the need for investments in transport infrastructure to ensure the competitiveness of commodities [11–13]. Therefore, reducing transportation costs is a pressing issue [14,15]. If, on the one hand, the growth of the world's GDP boosted Brazil's

economy due to the high demand for primary commodities, on the other hand, the country experienced severe problems regarding the lack of transportation infrastructure to offer competitive freight [11,15,16].

Furthermore, yearly soybean, corn, and iron records raised concerns about building an economically sustainable transportation infrastructure [17,18]. Nowadays, soybean freight accounts for 14% of the national volume of exportation; yet, it corresponds to 80% of the total transportation cost [9], with production reaching 135.4 million tons in 2020/2021 [19,20].

In order to maximize the competitiveness of the country in international agribusiness, Brazil released the National Logistic Plan (PNL) in 2012. Other than investments in existing ports, the government's response considered constructing and rehabilitating roads and railways to alleviate the impedance of cargo flow to the ports [16,21,22]. The PNL foresees a substantial attenuation of the transportation costs until 2035. However, the methods used to support and justify the assets are unclear [9]. Moreover, in practice, given the availability of the new transportation infrastructure, other than minimizing freight cost and travel time, alternative routes between the origin and destination are also expected [9,13,23,24].

Thus, a latent question still motivates our investigation: can PNL induce future port competitiveness by reshaping the port service areas? In this research, we hypothesize the role of the new infrastructure in changing the current service area of the ports, here called "macro-logistic basins", therefore driving port competitiveness never experienced in Brazil.

The objective of the investigation is to predict how the new transportation infrastructure proposed in PNL will rearrange the origin–destination matrix and reshape the macro-logistic basins of the Brazilian exporter ports. Therefore, we built a geographic-oriented model for predicting scenarios for routes and transportation costs, yet compiling and comparing the results of the current transportation infrastructure to the addition of new railway segments and highway rehabilitation proposed in the PNL. The model allowed us to:

1.  Critically analyze and question a potential imbalance between the expansion plans of the Brazilian transport infrastructure and the real demand for agricultural production that relies on the customs hegemony of geographically distant ports;
2.  Estimate the role of the proposed transport infrastructures in reducing freight costs;
3.  Assess the trend for changes in the service area of the ports traditionally used for exporting soybeans and corn, considering the implementations announced by the government until 2035.

In addition to delivering comparative maps and projections of existing national transportation infrastructure and a newly proposed one, this paper aims to provoke a discussion about port competitiveness and the potential for breaking Brazilian customs clearance and trade compliance paradigms. The document is organized as follows: Section 2 presents a historical aspect of the Brazilian National Transportation Plans that allows readers to understand the challenges of the national macro-logistics that respond to almost 1/3 of world soybean production. Next, Section 3 presents the research material and the methodology. Then, Section 4 presents the geographic-oriented model. Finally, Section 5 presents and discusses the results, and Section 6 concludes the study.

## 2. Spatiotemporal Contextualization of Brazilian Transportation Plans

Historically, Brazil has been one of the world's most important producers of agricultural, livestock, and mineral commodities since the 18th century [25,26]. Therefore, the sea and river ports were not only strategic for the economic establishment of the country but also the main driver of colonization and development of the major cities. At the end of the 19th century, some seaports became served by rail due to the increased volume of agriculture-based commodities [27]. Regarding the port infrastructure and the vessels that operate, the ports were built based on simple requirements such as the safety of calm waters strategically located in estuarine regions and bays [8]. By that time, the environ-

mental perspective and the impacts caused by the vessels were not a limitation for port operations [28].

From the Brazilian industrialization in the mid-1940s, the operational profile and structure of some critical ports underwent significant changes. Maritime ports such as Santos and Rio de Janeiro became even more strategic and boosted the attraction of international industries in their respective states [27]. On the other hand, other seaports such as Recife, Porto Alegre, and Salvador, as well as river ports such as Belem and Manaus, although very important in the structural composition of the country, did not offer competitiveness to the ports of Rio de Janeiro and Santos [8,27]. Nevertheless, the structure of the Brazilian customs clearance and trade compliance was set mainly to operate in these two ports, which directly and indirectly still exerts a strong influence on the processes that involve decision-making in the planning and operation of export cargo transport.

The politics initiated during President Washington Luis administration in the early 1920s induced the construction of roads and the proposed infrastructure to develop the country's interior [29,30]. The national interiorization plans were updated and remodeled during the administration of President Vargas (1930–1937), a period that included the creation of the National Department for Transport Infrastructure [31] and the National Department of Mineral Production, which was responsible for the regulation of these two highly correlated sectors, as well as the administration of President Kubitscheck (1956–1961), a period of installation of the first automotive industries [29]. Later, with the implementation of the National Transport System in 1973, the first national transportation act envisioned the integration of highways, railways, and waterways [32]. Thus, the national transportation policy encouraged agriculture's expansion toward the interior country.

It is also worth noting that EMBRAPA, a government agricultural research company, was also created in 1973 as part of the strategic planning of national economic development [ref]. EMBRAPA's strategy focused on training high-skill human resources alongside the installation of centers of excellence in research not only on the type of agriculture production in the region but mainly on the territorial vocation [26].

From this preamble, it becomes possible to understand the configuration of the national transport infrastructure, the spatial density of roads and railways to the east, and that of the central region by extensive agriculture experienced in the last 50 years. Figure 1 shows Brazil's current critical ports and the areas where agriculture and mineral commodities, the two pillars of the Brazilian GDP, originated for exportation.

In 2010, the Brazilian Federal Administration released the Transportation and Logistics National Plan (PNLT), a multibillionaire package for pushing transportation infrastructure projects, including the construction of thousands of kilometers of high-capacity railways, interstate highways, ports, and intermodal facilities [16]. The federal plan triggered successive political debates and disputes and, in 2012, was reformulated as the Integrated Logistics Plan (PIL), recently renamed National Logistics Plan (PNL) [9,22]. In essence, the PNL retains the nature of the precursor plans with no differences in infrastructure proposed but slight changes in transportation policies. For example, Figure 2 compares the existing transportation infrastructure (thin lines) to the proposed one from PNL 2035 (thick lines).

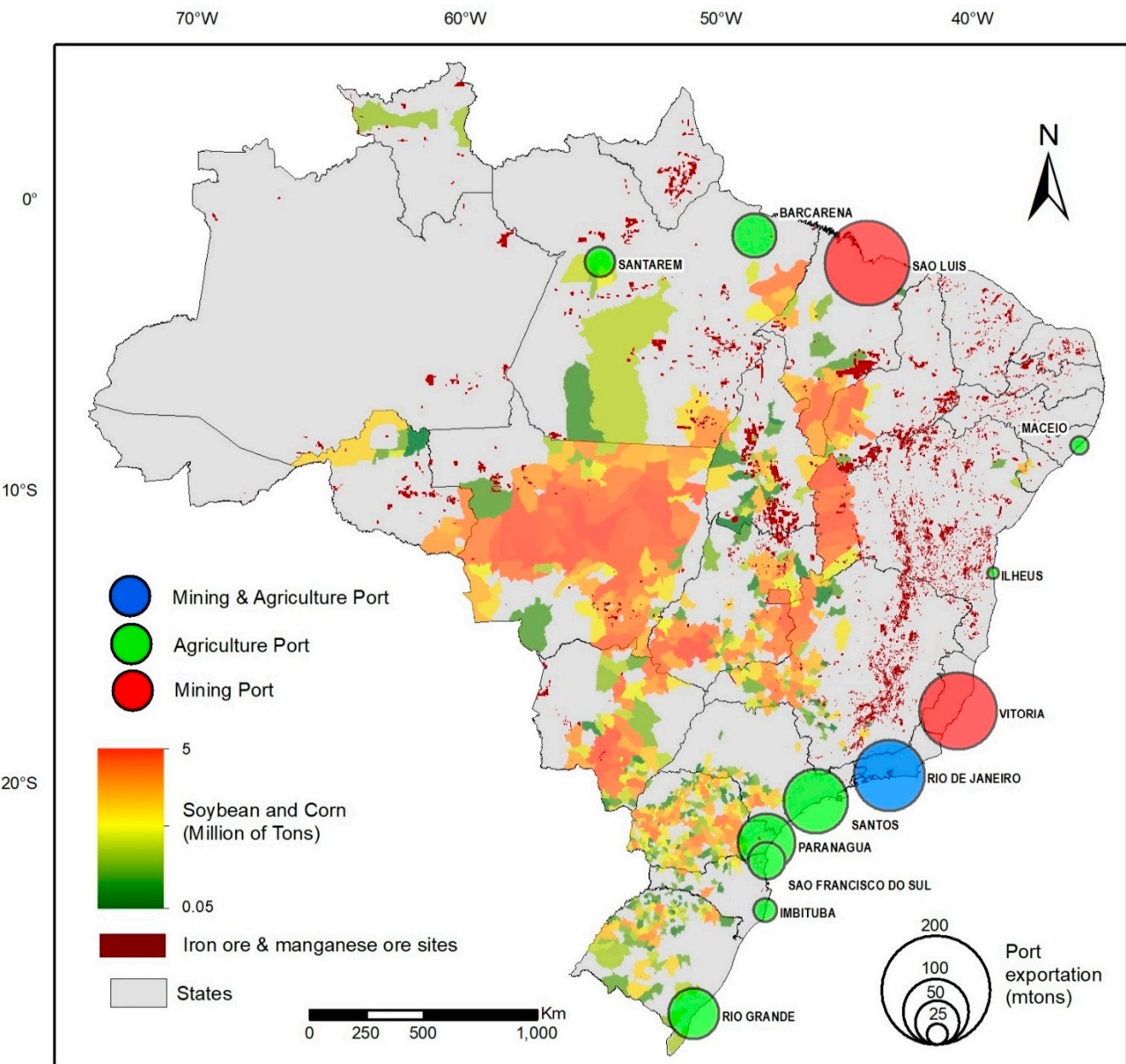

**Figure 1.** Brazilian ports and the origin of agriculture and mineral commodities.

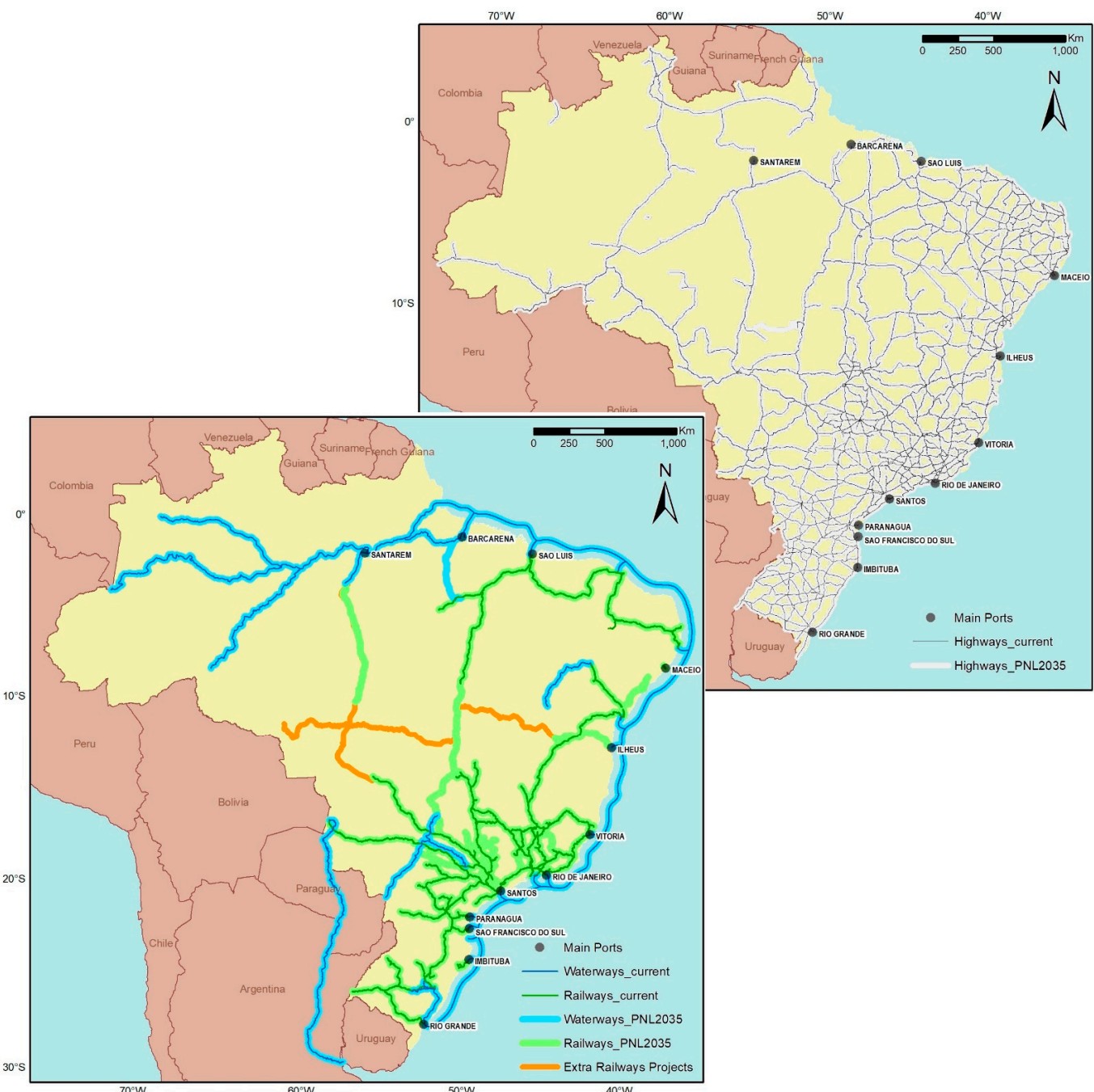

**Figure 2.** Brazilian current and planned 2035 transportation infrastructure.

## 3. Materials and Methods

### 3.1. Study Area, Data, and Tools

The study covered the 8.5 million km² of the conterminous Brazilian territory, including 3,246,787 km of roads, 33,849 km of rails, and 23,240 km of waterways [33,34]. All the materials used are official and publicly available, such as the Agriculture Gross Production by Municipality (AGPM) gathered from the Brazilian Bureau of Statistics and Geography (IBGE), and the geographic transportation database from the Brazilian Federal Company for Logistics and Planning (*Empresa de Planejamento e Logistica*—EPL), including the GIS layers of highways, railways, waterways, and ports [35]. In addition, the material included the Transportation and Logistics National Plan (PNLT 2010) and the National Logistic Plan (PNL 2035). Table 1 summarizes the input data and the respective sources.

**Table 1.** Input data, data format, and source.

| Input Data | Data Format | Source |
| --- | --- | --- |
| Existing Infrastructure: ports, railways, highways, waterways, and intermodal transfer nodes | Vector | Ministry of Transportation and Infrastructure (MINFRA) |
| Projected Infrastructure: railways, highways, waterways, and intermodal transfer nodes | Vector | Ministry of Transportation and Infrastructure (MINFRA) and Logistic Planning Company (EPL) |
| Municipality | Vector | Brazilian Bureau of Statistics and Geography (IBGE) |
| Agriculture Gross Production | Tabular | Brazilian Bureau of Statistics and Geography (IBGE) |
| Cargo transshipment Terminal and dry ports | Vector and Tabular | Logistic Planning Company (EPL) |
| Agriculture-based producer municipalities | Tabular | Brazilian Foreign Trade Statistics (Comex Stat) |

It is essential to highlight that mineral prospection commodities were not included in this study due to the dedicated railways connecting the plant straight to the port. Therefore, the well-established logistics for mineral commodities are not feasible for re-routing. In addition, similarly to the data, this investigation solely used free software. The data preparation and final maps composition used the QGIS version 3.24 Tisler [36], and the geographic model was developed using Dinamica EGO version 5 [37] and PostgreeSQL version 9.4 [38].

*3.2. Methodology*

To assess the effectiveness of PNL in reducing the transportation cost from farm to port, we created and compared two situations: Current Infrastructure and Current + Planned Infrastructure. Considering that the cost of the commodity is known both in its production and storage on the farm as well as on the shipment at the port, we considered transport cost as the simple difference in values.

Moreover, to investigate the hypothesis that changing the port's service area can drive port competitiveness, we added flexibility to the model by turning the ports into variables. In short, the farmer could ship the products to the nearby and cheaper port and receive goods such as fertilizers, fuel, and machinery. Thus, we computed two scenarios for each situation and then compared the results. The following major tasks summarize the procedures:

1. Create a GIS database with current transport, agriculture, and trading data;
2. Calculate the transportation cost and calibrate the model;
3. Add the layer of the PNL's planned infrastructure into the model and calculate the expected transportation cost;
4. Assess the least-cost routes from each municipality producer of soybean and corn origin to the ports;
5. Compute the ports' macro-logistics basins considering current and future scenarios;
6. Compare results and assess the effectiveness of areas of influence of the ports.

**4. The Geographic-Oriented Model**

The model is developed to compute the least-cost routes from the origin—municipalities that produce soybean and corn—to the exportation ports, and then to compute the service area of each port by connecting all the origins whose routes flow to the port. Figure 3 illustrates a schematic overview of the process.

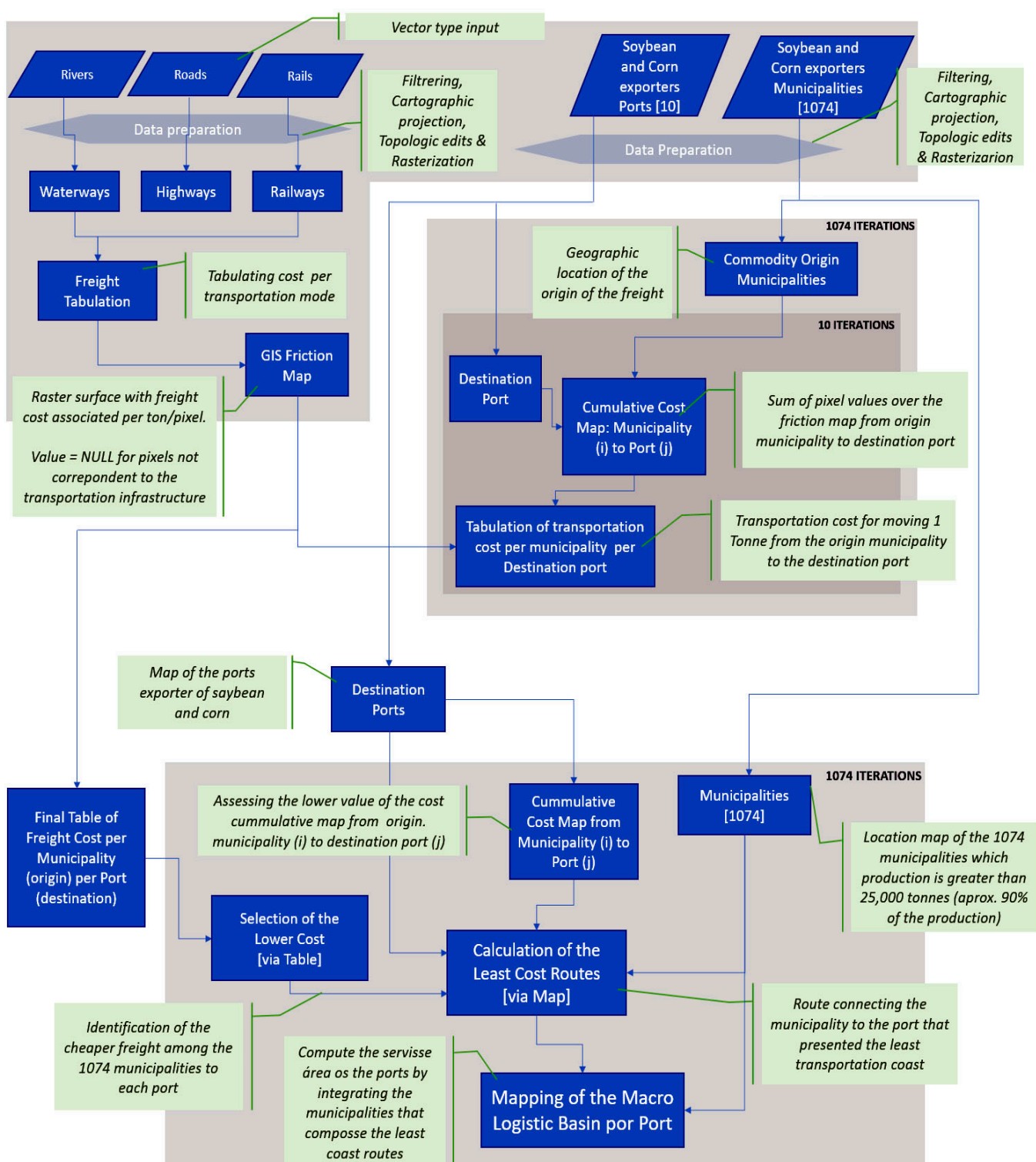

**Figure 3.** General flowchart of the model with comments.

This study uses simulation to search for the least cost to deliver the most likely path computed to best connect the origin (agriculture products) to destination ports. However, unlike many routing approaches that use discrete event solutions to model variables that instantaneously change the state at a specific time [23,39,40], this investigation is designed to provide an early assessment of future macro-logistics metrics resulting from the new proposed infrastructure. The goal is not to monitor the trip to optimize freight but to compute potential re-routing and changes in destination ports upon adding new planned

infrastructure. Consequently, our approach assumes that the route changes do not occur during the origin-destination trip. Thus, the location of the origin and destination points, the available transport infrastructure, and the cost per kilometer per type of mode used are the basic elements necessary for the solution.

The shortest path problem constitutes the largest group in the operational research area and is considered one of the most important in linear programming [9,41,42]. We use the well-known Dijkstra algorithm due to its robustness for solving complex topological problems and its availability on the PostgreSQL database via the *PG_Routing* extension [43–45]. Dijkstra's algorithm is a mathematical solution that calculates the minimum cost path between vertices of a graph [46]. It solves the problem of the shortest path in a directed or undirected graph as long as the weights of the connecting arcs have positive values. Choosing a vertex as the root of the search, this algorithm calculates the minimum cost of this vertex for all other vertices of the graph using a simple logic that delivers a good performance level [46,47].

Information gathered from EPL for each transport mode (road, rail, and waterway) [43,44] is used to calculate the cost of each section. The cost of each vertex represents the average freight cost per kilometer per ton. This information is available per transportation mode in the database of the EPL [48]. The PNL database includes the product sales invoice base as a reference for the price per ton at the origin and destination port. Thus, the costs of the soybean and corn on the farm (origin) and the soybean in the port (destination) are known. However, the cost of transportation and the routes are the variables to be modeled and estimated.

The transport infrastructure database provided by the EPL is not topologically ready for network analysis. Nevertheless, we use this database to feed the model inputs because they are the official public data containing both current transportation infrastructure and the one used by EPL to compute the scenarios presented in the PNL 2035. Figure 4 illustrates the database preparation for topological network analysis.

Regarding the data processing, the network topology is computed using PostGIS. In addition to the topological editing of the vectors, connections are also made from all points of origin (municipalities) and destination ports in the topological network. The duplicated sections are also removed, and segments with partial overlap are merged. Then, the length of the segments and the respective costs are computed according to the mode and type of pavement. The cost of each network segment represents the weight that the route optimization algorithm will consider. Because of the nature of the infrastructure data, the capacity of the planned infrastructure is set to maximum and is, therefore, not used as impedance. Dijkstra's algorithm is used to solve the weight of the segments regarding the freight cost and the length in kilometers [49].

The model operates on the Dinamica EGO platform through a logical instance that accesses PostgreSQL. The model's inputs are the list of origin municipalities, destination ports, and the type of commodity to compute the routes.

The routes are generated considering the least-cost path between origin and destination. The model draws routes over the transportation infrastructure layer. As a GIS vector layer, the attribute table of the route stores the amount of the product that traveled through the segment. This process is repeated for each OD pair referring to the selected municipality, generating a vector containing the originating routes and another containing the routes to the destination. The cumulative amount from the origin to the segment is also registered in the attribute table of each segment.

The validation aims to verify whether the model results reflect the results of the existing system [50]. We consider the database of the Statistical Yearbook of the National Agency for Aquatic Transport (ANTAQ), which records the movement of cargo in the Brazilian ports and data on total exports extracted from the EPL OD Matrix. We selected the seven main Brazilian ports with their maximum annual capacity, and the volume of Agricultural Solid Bulk (GSA) shipped from each port. Considering that the data on costs and transport infrastructure used by the EPL are not publicly available, we use the

infrastructure available at the Ministry of Infrastructure and the freight cost as the average freight cost per kilometer per ton, as well as information about cost from Agrolink [51]. Thus, the simulation result is expected to show that the volume of cargo shipped at each port is within the average reported by each data source. The results are satisfactory to validate the volume of cargo shipped in the main Brazilian ports as the ratio in percent of shipments from each port with the total of the observed set is average or close to it. In addition, the percent of the average of the ports concerning the total of each data source is also 14% (Table 2).

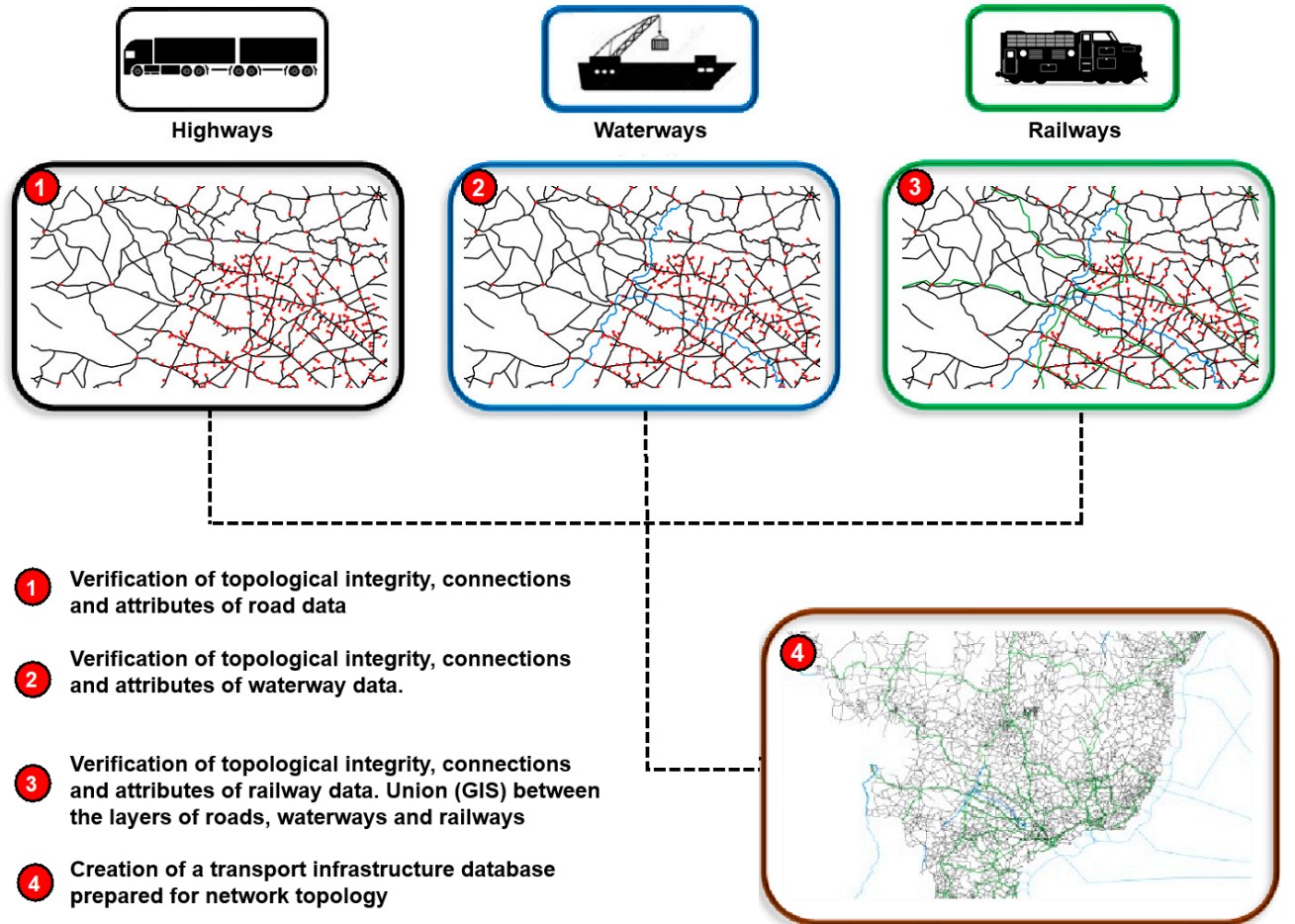

**Figure 4.** GIS database preparation for topological network analysis.

**Table 2.** Comparative metrics between the EPL and ANTAQ records and our simulation.

| PORT | CAPACITY (Ton) | EPL | | SHIPMENT (Tons)—2017 ANTAQ | | Simulated | | Average |
|---|---|---|---|---|---|---|---|---|
| Barcarena—PA | 20,000,000 | 15,062,183 | 14% | 16,231,030 | 13% | 22,106,339 | 15% | 14% |
| Santarém—PA | 5,000,000 | 9,880,068 | 9% | 4,293,581 | 4% | 8,074,788 | 6% | 6% |
| São Luiz—MA | 14,000,000 | 11,619,736 | 11% | 12,695,634 | 10% | 18,861,530 | 13% | 12% |
| Santos—SP | 70,000,000 | 35,313,821 | 34% | 51,235,967 | 42% | 54,651,517 | 38% | 38% |
| Paranaguá—PR | 35,000,000 | 12,991,638 | 12% | 18,765,173 | 15% | 22,217,292 | 16% | 14% |
| São Francisco do Sul—SC | 15,000,000 | 4,129,925 | 4% | 5,485,556 | 5% | 4,291,643 | 3% | 4% |
| Rio Grande—RS | 30,000,000 | 15,914,060 | 15% | 13,185,471 | 11% | 12,716,064 | 9% | 12% |
| Total | 189,000,000 | 104,911,431.96 | 100% | 121,892,412.56 | 100% | 142,919,172.20 | 100% | 100% |
| Average | **47,250,000** | **14,987,347.42** | **14%** | **17,413,201.79** | **14%** | **20,417,024.60** | **14%** | |

## 5. Results and Discussion

### 5.1. Transportation Cost Surfaces

In order to capture the nature of the geographic phenomena, the data processing for cost surfaces is developed in a raster format to ensure a better representation with

continuous variation in space. The raster format is also fully compatible with the data format used in the Dinamica EGO modeling platform. Figure 5 illustrates the transportation cost surface for the current infrastructure scenario (top) and the scenario considering the PNL-2035 infrastructure (bottom). The color ramp represents the variation in the cost of transportation from the local point to the nearest feasible exporter port, where red represents the high cost and green the low cost.

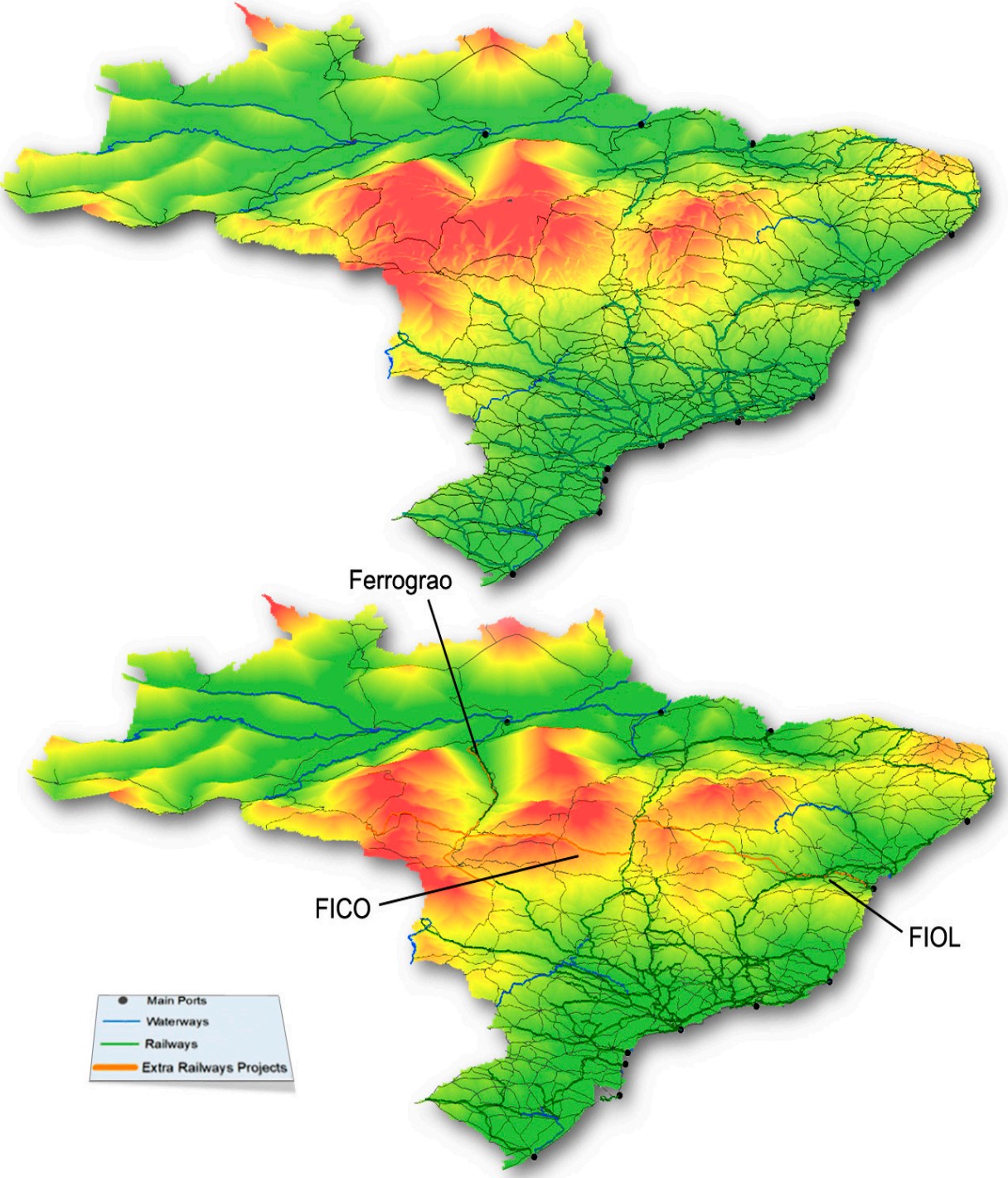

**Figure 5.** Perspective views. The geographically comprehensive cost transportation surface computed for soybean and corn considering the currently existing infrastructure (**top**) and the infrastructure proposed for 2035 (**bottom**).

As expected, findings from the cost surfaces computed from the current infrastructure scenario demonstrate that the highest transport cost is located in the most productive regions in the central region of Brazil. This result corroborates results from [9,10,13,23]. In addition to being the region furthest from ports, it is also the region least served by high-capacity and more efficient transport infrastructure.

However, when the PNL planned infrastructure is added to the model, the resulting transportation cost surface shows significant changes in the core production area in central Brazil. Metaphorically speaking, the railways Ferrogrão, FIOL, and FICO carve "valleys" across the transportation cost surface that will "drain" the agriculture production to the ports of Barcarena and Santarém at Madeira River and Atlantic ports of Ilhéus and Santos.

More specifically, the results reveal an effective reduction in transport costs resulting from the scenario that considers the FERROGRAO railroad, which aims to transport agricultural products from the state of Mato Grosso to the Port of Santarém and Barcarena in the north of the state of Pará. Currently, these two river ports depend exclusively on trucks that travel 1000 km along the precarious BR-163 road across the Amazon to load the ships.

### 5.2. Least-Cost Routes

The solution is developed so that, from vector inputs such as the map with the list of points of origin (producing municipalities) and points of destination (exporting ports), a map of the road, rail, and waterway network, and tabular information on the values of freight, the alternative routes are computed connecting each point of origin to the various points of destination, quantifying the corresponding freight values for each alternative and mapping the wheels of lower value. The geospatial approach operates in a blend of vector and raster architectures and seems to be an alternative for exploring both network topology and map algebra best. In addition to not being required in this study, the model also computes the least-cost routes departing from the port to each municipality. The solution will serve for future analysis regarding its use for modeling the freight return load shipped through the truck, train, and vessels. Figure 6 illustrates a sample of the double-way process.

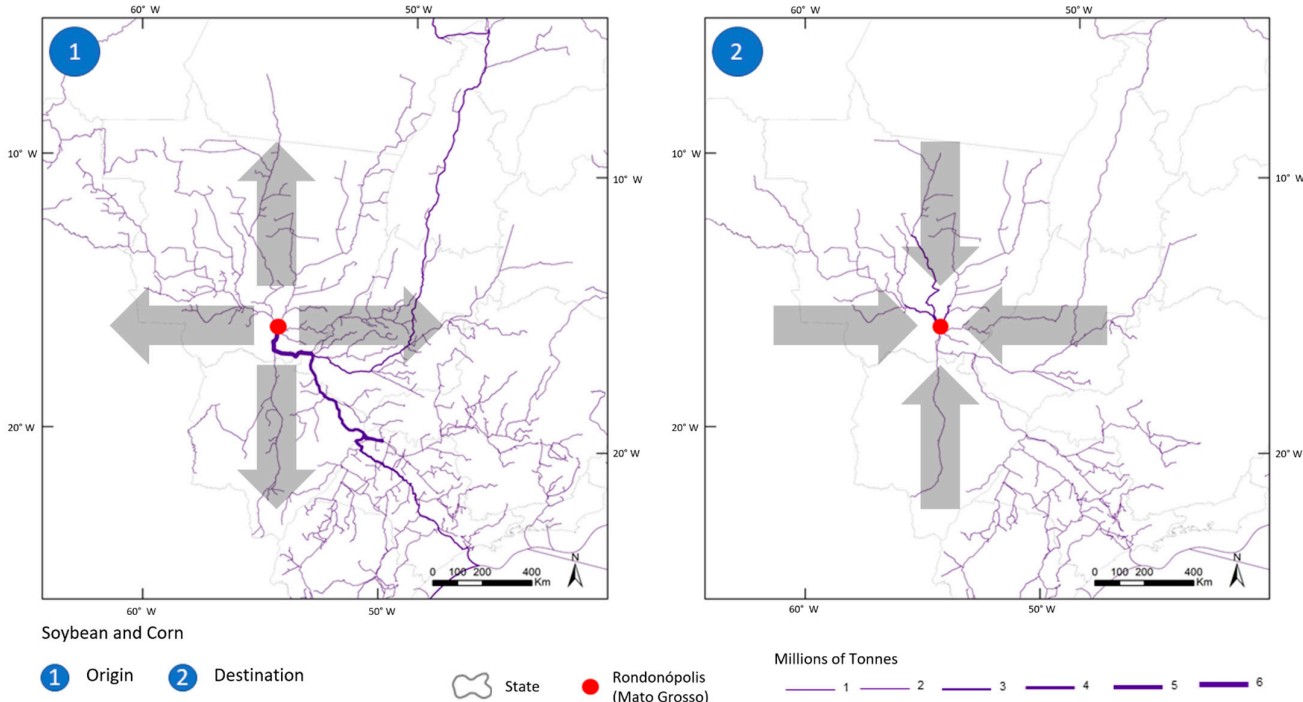

**Figure 6.** Example of a least-cost route computed from (**left**) and to (**right**) the municipality of Rondonópolis-Mato Grosso, Brazil.

### 5.3. Ports' Macro-Logistic Basins

In this study, we apply the EMBRAPA's concept for agricultural logistics basins [49], which relies on GIS tools to group the municipalities in which production flows preferentially through the same routes, modes of transport, and destinations. We compute the polygons of the macro-logistic basins considering the scenario of the ports influence area over the current transportation infrastructure and the scenario including the future PNL's infrastructure. The results are presented in Figure 7, top and bottom, respectively. Table 3 summarizes the volume of exportation per macro-logistic basin.

**Table 3.** Production of soybean and corn per macro-logistic basin.

| PORT | VOLUME OF EXPORTATION * (Tons) | | | |
|---|---|---|---|---|
| | **SOYBEAN** | | | |
| | **Current (Total)** | **Current (Exportation)** | **PLN 2035 (Total)** | **PNL 2035 (Exportation)** |
| Manaus | 5,718,194 | 3,341,918 | 2,599,399 | 1,721,800 |
| Barcarena | 7,497,700 | 6,292,440 | 7,832,583 | 6,627,323 |
| Santarem | 8,347,602 | 6,695,400 | 15,700,563 | 11,951,225 |
| Sao Luis | 1,296,246 | 975,327 | 1,296,246 | 975,327 |
| Vitoria | 193,131 | 185,351 | 193,131 | 185,351 |
| Santos | 29,443,282 | 24,079,018 | 24,885,336 | 20,119,532 |
| Paranagua | 17,483,741 | 14,313,816 | 17,483,741 | 14,313,816 |
| Imbituba | 101,636 | 101,636 | 101,636 | 101,636 |
| Sao Francisco do Sul | 2,322,522 | 2,181,002 | 2,322,522 | 2,181,002 |
| Rio Grande | 13,406,756 | 12,315,101 | 13,406,756 | 12,315,101 |
| TOTAL | 85,810,810 | 70,481,010 | 85,821,913 | 70,492,113 |
| | **CORN** | | | |
| Manaus | 1,025,846 | 2,924,338.10 | 481,452 | 1,359,920 |
| Barcarena | 665,275 | 2,242,976.29 | 698,668 | 2,338,464 |
| Santarem | 1,797,696 | 6,817,993.25 | 2,985,613 | 11,058,215 |
| Sao Luis | 67,691 | 691,411.13 | 67,691 | 691,411 |
| Vitoria | 33,161 | 700,481.00 | 33,161 | 700,481 |
| Santos | 3,555,472 | 11,307,406.13 | 2,878,556 | 8,536,114 |
| Paranagua | 1,359,201 | 2,828,491.42 | 1,359,201 | 2,828,491 |
| Imbituba | 484 | 460 | 484 | 460 |
| Sao Francisco do Sul | 6045 | 4560.70 | 6045 | 4561 |
| Rio Grande | 21,158 | 66,736.57 | 21,158 | 66,737 |
| TOTAL | 8,532,029 | 27,584,855 | 8,532,029 | 27,584,855 |
| | **SOYBEAN + CORN** | | | |
| Manaus | 6,744,040 | 6,266,257 | 3,080,851 | 3,081,720 |
| Barcarena | 8,162,975 | 8,535,417 | 8,531,251 | 8,965,788 |
| Santarem | 10,145,298 | 13,513,393 | 18,686,176 | 23,009,440 |
| Sao Luis | 1,363,937 | 1,666,739 | 1,363,937 | 1,666,739 |
| Vitoria | 226,292 | 885,832 | 226,292 | 885,832 |
| Santos | 32,998,754 | 35,386,424 | 27,763,892 | 28,655,646 |
| Paranagua | 18,842,942 | 17,142,307 | 18,842,942 | 17,142,307 |
| Imbituba | 102,120 | 102,096 | 102,120 | 102,096 |
| Sao Francisco do Sul | 2,328,567 | 2,185,563 | 2,328,567 | 2,185,563 |
| Rio Grande | 13,427,914 | 12,381,838 | 13,427,914 | 12,381,838 |
| TOTAL | 94,342,839 | 98,065,864 | 94,353,942 | 98,076,967 |

* Source of the production per municipality: SIDRA and ComexStat [26,35].

Generally, the Brazilian ports have a well-established area of influence. However, the consolidation of the area of influence relies not only on the services and the commodities whose destinations are foreign countries but also on the truck and train freight return loads, such as fertilizers, fuel, and equipment.

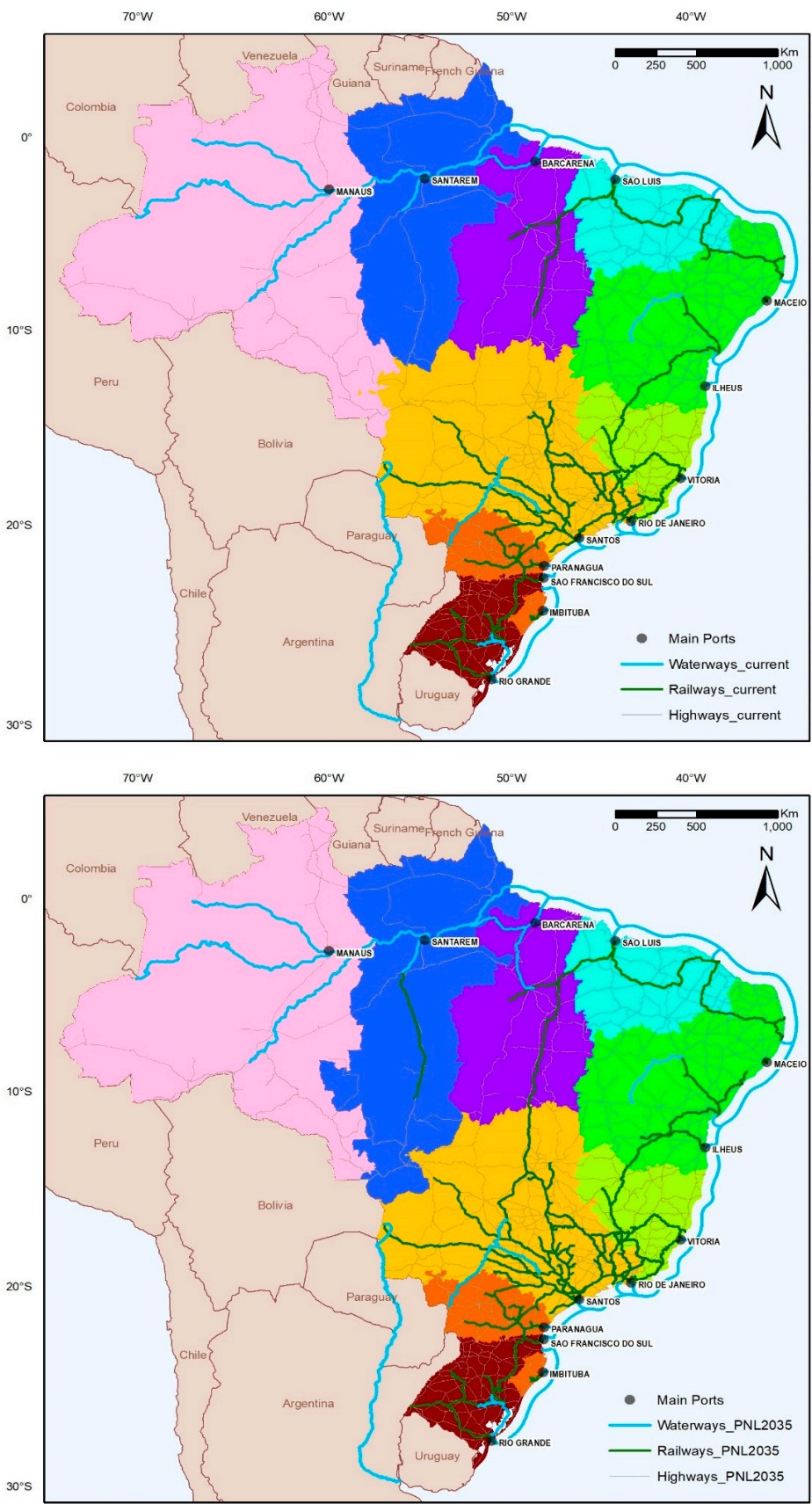

**Figure 7.** Macro-logistic basins and routes for soybean and corn computed using existing and planned transportation infrastructure.

With the relocation of the freight to the ports of the macro-logistics basin where they originate, we observe that shipment might represent a substantial reduction in the volume shipped to the Port of Santos and Port of Paranaguá and an increase in the ports of the Santarém and Barcarena in the state of Pará. This reallocation also indicates that there is no need for new corridors for agricultural production flow in the Midwest region, and a reorganization of the transport system might be enough. However, investments are necessary for ports that will increase the operation volume.

### 5.4. Changes and Trends in the State of Mato Grosso

For over a decade, the state of Mato Grosso surpassed records in agriculture production [20,41,52]. However, its geographic location remains a challenge for the national logistic [24]. Mato Grosso is located in the geographic center of South America, farthest away from any existing port in the country. If, on the one hand, the production cost of the Brazilian soybean is relatively low compared to the USA, the long distance to reach the ports, as previously shown in Figure 1, and the transportation infrastructure significantly increases the commodity's final price. Therefore, in this study, we believe that the most significant changes in the area of influence of the ports can be predicted for the state of Mato Grosso. Figure 8 highlights the changes.

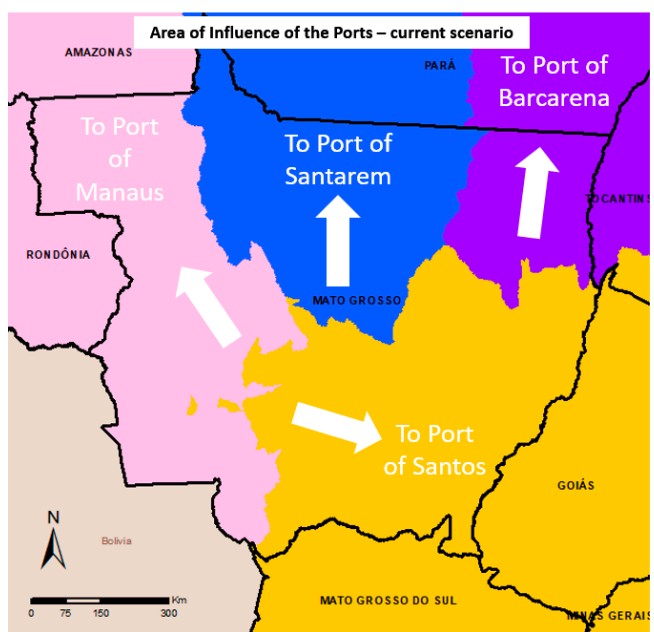 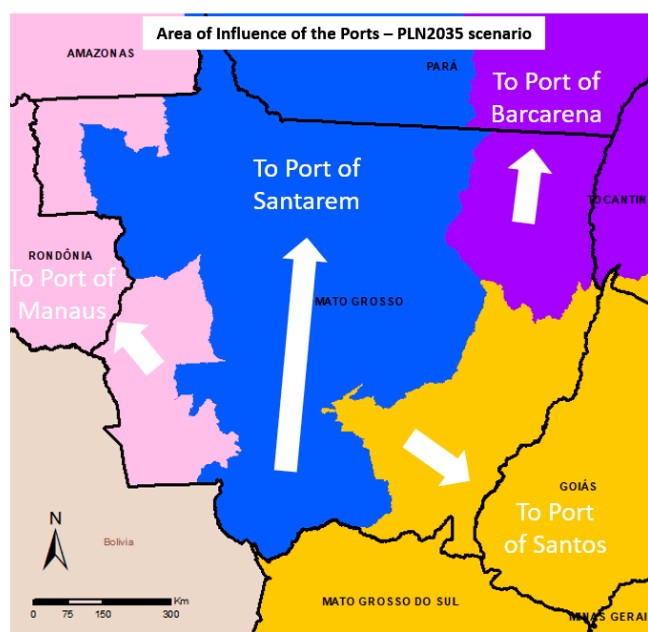

**Figure 8.** Predicted changes in the macro-logistic basins after the new transportation infrastructure proposed on PNL 2035 for the state of Mato Grosso in Brazil.

It is also possible to establish the average transport cost by state and the reduction caused by implementing the new corridor. Therefore, only the states of Pará and Mato Grosso would benefit from the implementation of Ferrogrão, and the FICO and BR-242 highway corridors, as proposed on PNL 2035, do not drive significant changes in the cost of transport to the states. Considering the region of the state of Mato Grosso as the largest producer of grains and where transport costs are higher, among the projects studied, implementing the Ferrogrão railway is the most significant cost reduction (as shown in the valley carved on the cost surface, Figure 5).

Regarding the port infrastructure in the process of expanding the Brazilian road and rail networks for 2035, it is important to highlight that the physical and legal limitations of the ports were not properly observed in the PNL. This gap can significantly compromise the success of the plan. Environmental issues already limit the port operation in many of the traditional Brazilian ports and the massive urbanization of its neighborhood. In

addition, the physical aspects, such as the lack of a deep draft, limit the ports to operate vessels compatible with the new panamax and chinamax standards. For example, Santos and Paranaguá (Figure 9), two of the most important ports in the country used to export agricultural commodities and import fertilizers, do not enable simple expansions due to socioenvironmental legislation and physical constraints.

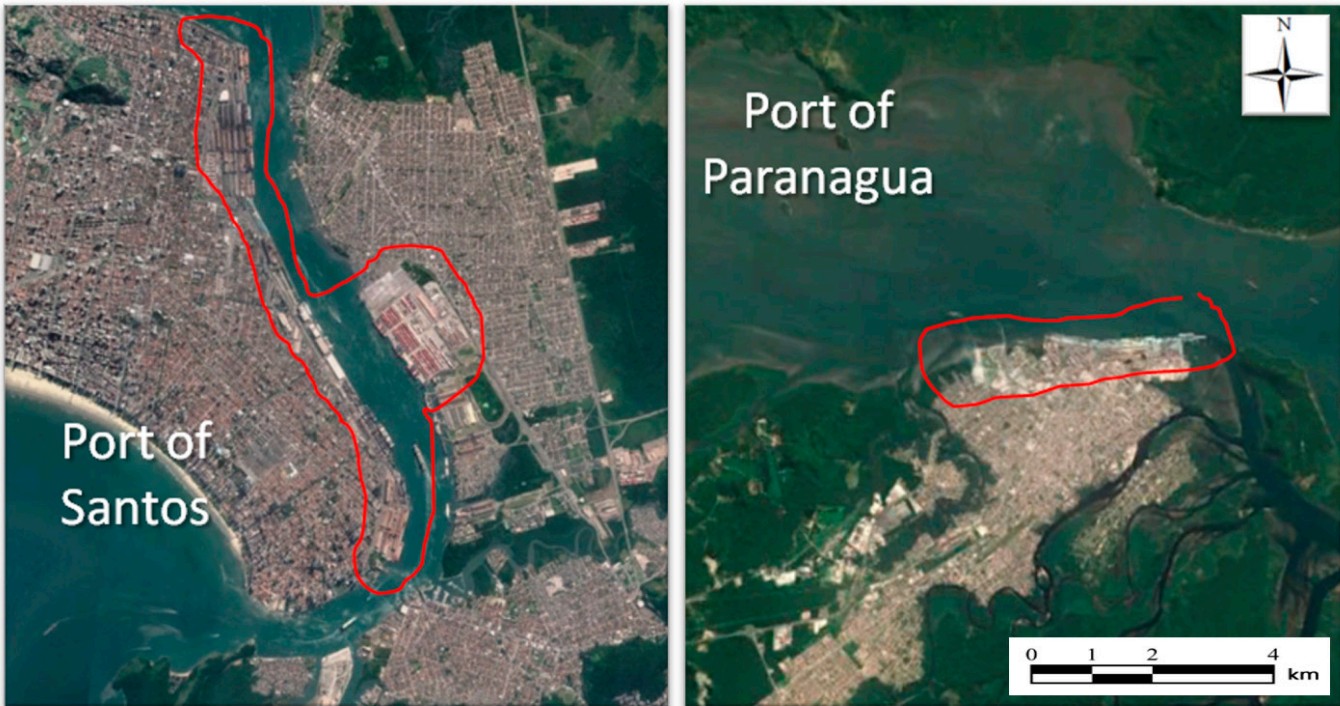

**Figure 9.** Aerial view of the vicinity of the three ports responsible for exporting 70% of the grain from Brazil. Background: Google Earth.

## 6. Conclusions

The answer to the question that motivated this investigation is yes; the proposed PNL-2035 can induce port competitiveness by reshaping the port service areas. The grain-producing regions in Brazil are very far from the ports where such commodities are shipped to the international market. There is no real competition among the ports, because the lack of comprehensive transportation infrastructure does not offer feasible alternatives to flow agricultural commodities to the nearby port.

The model developed in this research fulfilled its objective by providing mechanisms for geographic modeling and geoprocessing designed to support the spatial understanding of the nexus between the agricultural production chain and macro-logistics for export. Furthermore, the model is replicable and developed using free software and publicly available data.

The results obtained with the model fed with the infrastructure planned for 2035 prove that there is a tendency for agricultural commodity loads (export) and inputs (import) to be transferred to operation in the nearest ports where transport costs are lower, considering that whoever pays for the freight is the producer. The "greenfield" projects foreseen in the PNL 2035, such as the FICO, FIOL, and Ferrogrão railways, will not only reduce the cost of transport but also expand the area of influence of the ports directly served by these railways.

It is notable that, considering that the geographical configuration of Brazilian ports admittedly does not favor the logistics of transporting agricultural products, and considering that Brazil is breaking records on agriculture-based commodities, one of the observations that guided this investigation was: "when will the transportation planners and decision-makers realize the need to redistribute cargo between ports?". Thus, these geographic

adjustments quantified with the model will drive a significant amount of freight of agricultural commodities from the state of Mato Grosso straight to the ports of Barcarena and Santarém at Madeira River. Changes in the geographic domain of the traditional Brazilian southeastern ports is certainly motivating port competition, however what is not clear yet is the degree of the economic impact.

**Author Contributions:** Conceptualization, W.C., B.S.-F., and R.N.; Methodological Design, W.C., B.S.-F. and R.N.; Software, W.C.; Data Processing, W.C.; Formal Analysis, W.C. and R.N.; Validation W.C.; Writing, R.N.; Funding, B.S.-F. All authors have read and agreed to the published version of the manuscript.

**Funding:** This research was funded by the Remote Sensing Center of the Federal University of Minas Gerais, and by the National Council for Scientific and Technological, Process # 315631/2021-0 and # 312433/2018-2.

**Institutional Review Board Statement:** Not applicable.

**Informed Consent Statement:** Not applicable.

**Data Availability Statement:** Not applicable.

**Acknowledgments:** We would like to thank the Conselho Nacional de Desenvolvimento Científico e Tecnológico (CNPq) for funding R.N. research during this project.

**Conflicts of Interest:** The authors declare no conflict of interest.

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
