# Peer review of "Can the Brazilian National Logistics Plan Induce Port Competitiveness by Reshaping the Port Service Areas?"

_sustainability, doi:10.3390/su142114567_

Round 1

Reviewer 1 Report

Thanks for submitting your paper to Sustainability. This paper aims to evaluate the efficiency and effectiveness of these new projects. Overall, the paper’s quality is adequate, and topic is interesting. Please find my comments below.

-      In the abstract, there are no implications/value of this paper. Please add it. Also, the research objective is not clear. Currently, the authors says that “this research aims to evaluate the efficiency and effectiveness of these new projects”. Please be more specific.

-      There are no literature review section. Without this, it is difficult to identify why the current study is unique. Introduction does not effectively show comprehensive prior studies regarding this topic. Introduction just focused on Brazilian logistics contexts. Therefore, please add literature review sections. Also, please add some important prior studies how ports can improve regional development (e.g. (1) The impact of seaports on the regional economies in South Korea: Panel evidence from the augmented Solow model, Transportation Research Part E: Logistics and Transportation Review 85, 107-119, (2) The role of seaports in regional employment: evidence from South Korea, Regional Studies 52 (1), 80-92).

-      Table 4 looks blur. Please change it with higher resolution.

-      Methodology and results parts look robust, but I have concerns in conclusions. The current conclusion only shows some results, so it looks like an abstract. It lacks theoretical/managerial/policy implications for policy makers of governments. Please add some.  

Reviewer 2 Report

This paper puts forward a geographical model to evaluate the impact of the New-Proposed Infrastructure on the competitiveness of ports. The study is interesting and valuable conclusions are drawn. The manuscript has a certain dose of intrinsic merit but is not publishable as it stands. I have several comments which may help the authors to further improve the quality for publication standard.

1. The introduction, though industrious, fails to highlight one very relevant aspect, namely the original contributions of the manuscript. The authors hint at these in other sections but should offer a clear discussion in this area.

2. The way of expression is not rigorous. The poor logicality between the research results and the conclusions leads to the formation of a general management revelation with low credibility.

3. There are few references, and the related literature review is rather dull, so it is difficult to highlight the theoretical value of this paper.

4. The resolution of Figures are too low for me to see clearly. Please improve the sharpness of the picture. In addition, I have not seen any introduction about fig. 4.

5. Any limit?

Reviewer 3 Report

The research presented in the paper is interesting. But at this moment some major shortcomings can be identified. The first of these is represented by the mathematical model used that is missing from the presentation. The scenarios used are not clearly presented, nor is the way in which the model is calibrated (not validated). Also, the degree of confidence is not presented for the results obtained.

The graphic representations are diffuse and it is not clear what each one is used for (see figure 2). 

I am waiting for a second version of the paper in which the models used are better explained.

Round 2

Reviewer 1 Report

I think that the authors put much effort to revise the paper. Now, paper's quality is much improved, so I recommend 'accept'. 

Author Response

Thank you much for your kind attention and excellent suggestions.  Please find attached the modification we made in this new manuscript version. 

Reviewer 3 Report

The authors made a series of changes without taking into account my observations. The paper is interesting, but it is still unclear to me how the calibration and validation process was carried out and what is the degree of confidence. The authors have limited knowledge regarding the operation of the transport infrastructure. Using a minimum path algorithm without taking into account capacity limitations, transfer nodes, involved operators is not correct. I have the kindly request for them to improve this part of the research.

Author Response

(The authors gave the same response as above.)
